# Flow Neural Network for Traffic Flow Modelling in IP Networks

## Abstract

This paper presents and investigates a novel and timely application domain for deep learning: sub-second traffic flow modelling in IP networks. Traffic flows are the most fundamental components in an IP based networking system. The accurate modelling of the generative patterns of these flows is crucial for many practical network applications. However, the high nonlinearity and dynamics of both the traffic and network conditions make this task challenging, particularly at the time granularity of sub-second. In this paper, we cast this problem as a representation learning task to model the intricate patterns in data traffic according to the IP network structure and working mechanism. Accordingly, we propose a customized *Flow Neural Network*, which works in a self-supervised way to extract the domain-specific data correlations. We report the state-of-the-art performances on both synthetic and realistic traffic patterns on multiple practical network applications, which provides a good testament to the strength of our approach.

## 1 Introduction

Deep Learning (DL) has gained substantial popularity in light of its applicability to real-world tasks across computer vision, natural language processing (Goodfellow et al., 2016), protein structure prediction (Senior et al., 2020) and challenging games such as Go (Silver et al., 2017). Typically, the data for these learning tasks takes the form of either grids, sequences, graphs or their combinations. The tremendous efforts on customizing neural network structures (Krizhevsky et al., 2012; Kiros et al., 2015; Hochreiter & Schmidhuber, 1997) and learning strategies (Sermanet et al., 2018; Oord et al., 2019) to explore the data-specific properties underpin the success of modern DL in these domains. Following the same design philosophy, we wish to capitalize on these advancements to develop a customized neural network and self-supervised learning strategy to tackle the crucial and timely challenge of traffic flow modelling in IP networks.

### 1.1 Traffic Flow Modelling in IP Networks

An IP network is a communication network that uses Internet Protocol (IP) to send and receive messages between one or more devices such as computers, mobile phones. The messages could be general application data such as video, emails or control signals of any connected devices. When sending the messages from a source to a destination, the source device encapsulates the bit chunks of encoded messages into a set of IP packets. The packets then travel through communications links and routers or switches in a given routing path sequentially, thus forming the traffic flows in an IP network (Hunt, 1992). As one of the most commonly used global networks, the IP network provides the majority of such data transmission services to support today's Internet applications such as video streaming, voice-over-IP, and Internet of Things. Therefore, a good understanding of the behaviorial patterns of the underlying traffic flows plays a crucial role in network planning, traffic management, as well as optimizing Quality of Service (QoS, e.g., transmission rate, delay). This challenge is termed as traffic flow modelling and is fundamental to IP networking research and practice. However, the high nonlinearity, randomness and complicated self similarity (Leland et al., 1994) of these traffic thwart extensive traditional analytical and learning models, particularly at fine-grained time scales, such as traffic flow modelling at a sub-second level.

Consider the illustrative example in Fig. 1, which depicts multiple packet flows with shared forwarding nodes and links in their routing paths. The sender of each flows streams data packets to the

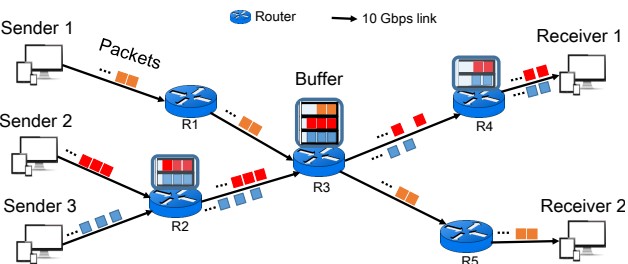

Figure 1: Traffic flows in IP networks.

receiver at a dynamic sending rate, which is determined according to many factors such as its rate demand, existing traffic loads, available link bandwidth, and etc. The packets usually experience various delays on the journey due to actions such as forwarding processing, link transmission, packet queueing. For example, when the sum rate of Sender 2 and 3 exceeds 10 Gbps, the router R2–R4 will hold off and cache the arriving packets in their buffers until the links from R2 to Receiver 1 become free, causing what is known as the queueing delay. The extent of these delays depends on multiple factors, including the amount of traffic going on, the capacity of the router's output queue, link bandwidth etc. The random establishment, interaction and termination of massive flow connections give rise to network dynamics. This illustrates the complexity of traffic flow modelling in IP network even for the simple example. This challenge is exacerbated when the traffic loads are running at over 100 Gbps and in a network with significantly larger size in practice.

## 1.2 MOTIVATING FLOWNN BASED TRAFFIC FLOW MODELLING

A flow pattern can be defined as anything that follows a trend and exhibits some kind of regularity, e.g., distribution, periodicity etc. The modelling of traffic flow patterns can be done mathematically or by the use of data-driven learning algorithms. We argue that developing a customized *FlowNN* in the context of IP traffic flow modelling is important in two aspects: 1) improving the performances of supported network applications from the accurate modelling towards the behavioral patterns of traffic flows in IP network, particularly at the time scale of sub-second level; 2) providing an exciting new "playground" and neural network model for the DL community to solve real-world-motivated research challenges by deeply combining its structure and working mechanisms. Next, we make the following two clarifications.

**Why not using traditional mathematical models.** The past decades have seen numerous traffic models proposed to mathematically model the traffic characteristics of networks (Gebali, 2015). For example, extensive studies use the Poisson model to characterize the traffic by assuming the arrival pattern between two successive packets follows Poisson process. Considering the heavy tailed distribution and burstiness of the data-center traffic, recent work in Benson et al. (2010) models the traffic arrival pattern as a log-normal process. To capture the temporal patterns and make predictions accordingly, Seasonal Autoregressive Integrated Moving Average (SARIMA) is exploited in (Ergenc & Ertan, 2019) to model the traffic time series. These analytical models may generate outputs that are easier to interpret, but are bonded to the specific working circumstance and assumptions. More importantly, these statistical models function at coarse time scales of hours and assume relatively smoother traffic patterns. However, as reported in many practical traffic measurements in e.g. Benson et al. (2010; 2011); Greenberg et al. (2009), most flows last less than 1 minute. This implicates tasks requiring traffic models at finer-grained time scales are beyond the capability of these traditional models.

Fig. 2 plots the traffic traces we collected from a practical backbone network–WIDE[1], which shows the *realistic* traffic patterns when the packet flows are sampled by two different time scales. The long time-scale plot in Fig. 2b shows clear a "tide-effect" associated with daily human activities. By contrast, the traffic traces in Fig. 2a get more noisy and difficult to recognize obvious patterns when they are counted by millisecond.

---

[1] http://mawi.wide.ad.jp/~agurim/index.html

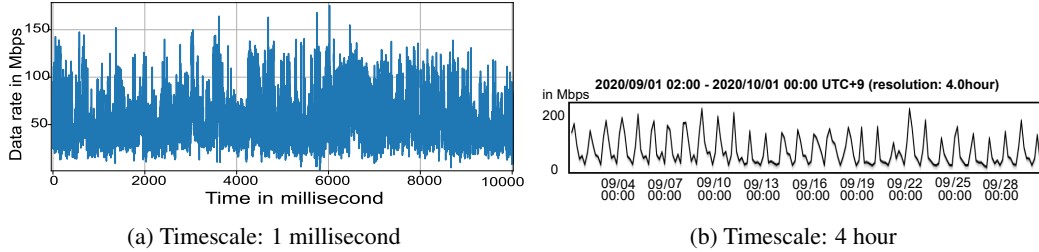

(a) Timescale: 1 millisecond          (b) Timescale: 4 hour

Figure 2: Time series of practical traffic traces from WIDE backbone network binned by two different time scales.

**Why not using existing neural network models.** When put in the context of data-driven learning, the traffic flow modelling problem can be reduced to the representation learning task. If treating the traffic flows as the general spatio-temporal data, extensive existing neural networks fit such task, including Convolutional Neural Net (CNN, (Mozo et al., 2018)), Graph Neural Net (GNN, (Rusek et al., 2019)), Recurrent Neural Net (RNN) as well as their variants and combinations (e.g., STHGC-N (Kalander et al., 2020), STGCN (Yu et al., 2018), and Xiao et al. (2018); Polson & Sokolov (2017); Cui et al. (2020); Guo et al. (2019); Lin et al. (2019)). The customized designs to the data-specific properties make the success of these existing models, such as the convolutional operation to capture the spatially local correlations in CNN and the aggregation operation to extract the adjacent link correlations in GNN, and so on. As a human-engineered industrial system with clear system structure and working mechanism, the IP network creates domain-specific spatio-temporal data correlations, which are difficult to capture for the incumbent spatio-temporal models if without any modification. One of the most important difference is that the spatial data in IP networks is not only correlated to other spatial data at the same point in time, but also able to directly influence the future realizations of correlated locations with strict order (i.e., the *Spatio-Temporal Induction* effect as we will disclose later). Moreover, these existing studies only target at a coarse-grained timescale above minutes or even hours. Models at a sub-second granularity, as *FlowNN* functions, require deeply combining the spatio-temporal data trends, as well as the system structural knowledge and working mechanism.

## 1.3 Our Contributions

We claim two critical contributions: 1) we formulate the crucial traffic flow modelling problem in IP networks as the representation learning task in deep learning, and develop a customized neural network–*FlowNN* and the associated *Induction Operation* to extract the domain-specific spatio-temporal data correlations in IP traffic flows. To the best of our knowledge, this is the first work to design customized neural network and learning strategy by deeply combining the IP network structure and working mechanism. The *Induction Operation* also makes it the first time for a data-driven learning model able to infer the data features at a millisecond granularity, which is usually treated as the '*noise*' region by existing coarse-grained models; 2) we report the state-of-the-art performance over the baselines in different type of practical network applications, which provides a good testament of our model.

## 2 Spatio-Temporal Induction

By stacking the networking feature timeseries sampled at all the nodes that a flow passes through, the IP traffic flow data can be organized in the form of high-dimensional tensor time series, as shown in Fig. 3a. The feature (denoted as $x_{f,n}^t$) could be the average flow rate (i.e., the amount of packet bits received in each unit measurement time) at each node or the average per-hop packet delay etc. The routing path herein constitutes the most significant attribute for the generative process of each IP traffic flow. This creates many peculiar properties in such flow data. For example, for a flow with a routing path [1→4→12] in Fig. 3a, the current data at node 4 was originated from the history data at its predecessor node 1, but delayed by at least[2] the link delay $\Delta_t$. These data will also *flow* to its successor node after certain delays. This shows that the flow state at a node is *physically*

---

[2]Packet processing and queueing will also impose extra delay.

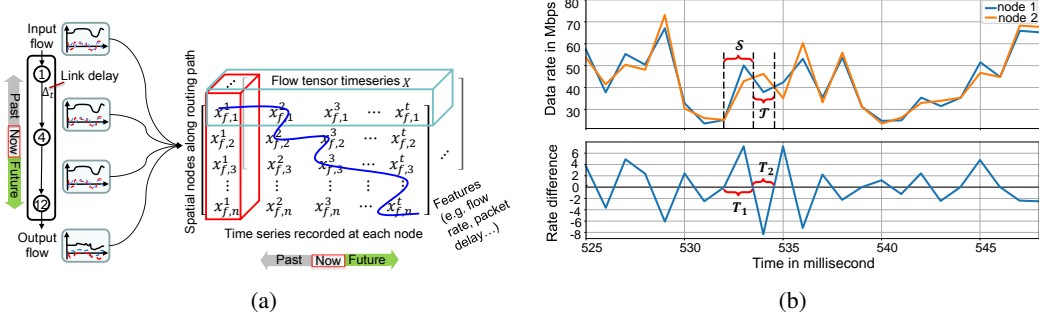

Figure 3: (a) IP traffic data in the form of flow tensor time series. The S-shaped curve characterizes the data correlationship along the spatio-temporal dimensions; (b) A sample of data rates of a flow and the rate difference recorded at two neighboring nodes.

driven[3] by the past flow state at its predecessor node. Therefore, such time-resolved flow data not only tells us *who* is related to whom but also *when* and in which *order* relations occur. This forms the S-shaped data correlation pattern, as exemplified in Fig. 3a. An accurate modelling of the flow data requires attention to such domain-specific data properties, which are absent in the majority of existing learning models if not all of them.

Fig. 3b plots a sample of flow traces at two neighboring path nodes from the WIDE dataset. We can observe that an exceeding of data rate at node 1 over node 2 for some duration (e.g., $T_1$) will always *induce* a subsequent duration $T_2$ when the data rate at node 2 is greater than that at node 1. Moreover, the cumulative amount of data bits the two nodes forward in such two durations are almost same, as indicated by the rate difference between the two nodes at the bottom of Fig. 3b. This illustrates that there is a stronger correlation among the data in these two durations, and the future realizations at $T_2$ are subject to the constraint of the states at $T_1$ by local flow conservation[4] between the two nodes.

In analogy to the concept of *Electromagnetic Induction*[5] in Physics, in what follows, we introduce the *Spatio-Temporal Induction (STI) effect* in IP traffic flows. Accordingly, a learning model is proposed to model the network traffic flows at a fine-grained timescale.

**Definition 1.** *Spatio-Temporal Induction is the production of the temporal evolutions of a flow from the history spatial patterns at its correlated locations.*

The STI builds a physical relationship between the spatial and temporal patterns of IP flow data. This provides a more accurate interpretation to the underlying data generating process than the trending information manifested by the data itself. Such induction effect is created by the IP network structure and working mechanism, which is preserved when the flow data is sampled at any timescale in practice.

Next, we develop an *Induction Operation* to concretely capture the S-shaped data correlation in IP flow data and propose what we call *FlowNN* to generate the desired traffic model for IP networks.

## 3 FLOW NEURAL NETWORK

As aforementioned, after the packet flows leave the source devices, they may experience delays from many factors (e.g., link transmission, packet processing, queueing etc) during the journey throughout the routing path. Accordingly, each node in the path will record a transformed time-series view of the same source flow due to the transmission and forwarding actions from the network. This

---

[3]The exotic competition from other co-located flows also drives the flow evolution in part. This can be included by appending the aggregated coflows as an auxiliary feature.

[4]Flow conservation is the network property that except the source and sink, the total incoming flow of a node is same as its outgoing flow or incoming flow of its successor node in the routing path if no packet is lost.

[5]*Electromagnetic induction* is the production of electric currents across a conductor in a changing magnetic field, which determines a relationship between electricity and magnetism.

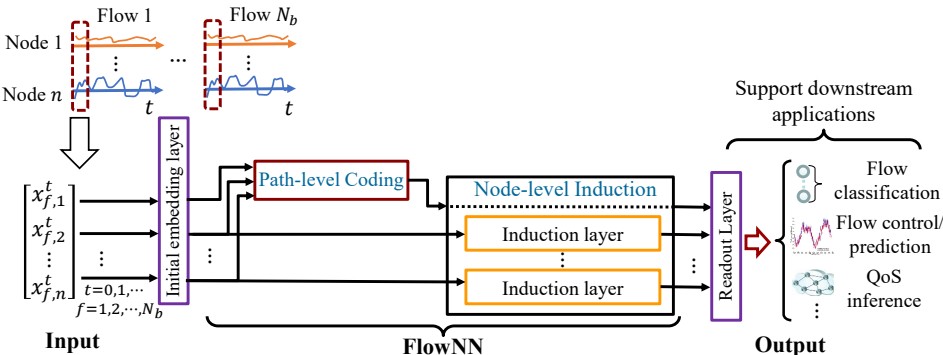

Figure 4: Framework of *FlowNN*. With the raw feature data $x_{f,n}^t$ as inputs, *FlowNN* extracts both the path-level and the induced node-level feature patterns as learned feature vectors, which can then be used for different application tasks by changing the Readout layer.

implicates the existence of similarity among these time series since they are from the same source flow. Such path-wide similarity reflects the shared forwarding policies all path nodes follow, as well as the original behaviors of the source flows at the sending side, such as the dynamics of user demands. On the other hand, the flow patterns recorded at each node may end up being different from the patterns at the sending side, depending on the experienced networking actions. With all these in mind, we attend to such similarities and differences by two key blocks–*Path-level Coding* and *Node-level Induction*, and build the framework of *FlowNN* as Fig. 4.

***Path-level Coding:*** This block constructs an encoding network to extract the patterns that can represent the state of the whole path. Such path-level patterns capture the similar part of behaviors within the time series received from all path nodes and will be the base to derive the evolved patterns at each path node. Technically, we tested both the very recent Siamese net (Maron et al., 2020; Ye et al., 2019) and GNN based encoders (Rusek et al., 2019) to extract the path-level patterns. Surprisingly, they are not outperforming a direct encoding by the Gated Recurrent Unit (GRU) with all raw node data as inputs. This stands to the reason that the raw data from the correlated locations conveys the direct guidance for the near future realization at a node. The operations in above models, e.g., SumPooling or node aggregation, actually destroy such data correlations and introduce more 'noise'. More details are provided in later experiments. Consequently, in the following, we directly apply a GRU to encode all the raw inputs.

***Node-level Induction:*** This block encodes the individual patterns at different nodes conditioned on the path-level patterns from the source flows. Looking into how flows are induced in Fig. 3b, we can observe an explicit data correlations subject to local flow conservation among the correlated data sequences. Next, we propose the *Induction Operation* and the associated *Contrastive Induction Learning* to extract such data correlations.

As illustrated in Fig. 3b, the induction process operates on any two consecutive durations like $T_1$ and $T_2$, which can be identified through the rate differences between two neighboring nodes. The induction is performed separately for each path node, as shown in Fig. 4. For ease of exposition, we denote the data sequences at all path nodes in $T_1$ and the data sequence at the induced node in $T_2$ as $\mathcal{S}$ and $\mathcal{T}$, respectively. Then, the *Induction Operation* performs the conditional induction function $f(\mathcal{T}|\mathcal{S})$ to capture the correlations between source sequence $\mathcal{S}$ and target sequence $\mathcal{T}$. The function $f(\mathcal{T}|\mathcal{S})$ can be approximated by a specialized *Contrastive Induction Learning* in a *self-supervised* manner as follows.

***Contrastive Induction Learning:*** As discussed in ***Path-level Coding***, we first take the outputs of path-level feature patterns, $\{h_{\mathcal{S}}^t\}_{t \in T_1}$, as the initial coding states of source sequence $\mathcal{S}$. A recurrent encoder (e.g., GRU[6] (Chung et al., 2014)) is then applied to generate the context $c$ of $\mathcal{S}$ with $\{h_{\mathcal{S}}^t\}_{t \in T_1}$ as inputs. Conditioned on the context $c$, a GRU decoder can be then used to produce the

---

[6]Note that the encoder is not limited to GRU, but any type of recurrent models can be used. More recent recurrent models stacked with modules e.g., attention blocks could help improve results further.

state codes, $\{\hat{h}_{\mathcal{T}}^t\}_{t \in T_2}$, for the target sequence $\mathcal{T}$. With the state codes of both $\mathcal{S}$ and $\mathcal{T}$ at hand, the key of the induction function $f(\mathcal{T}|\mathcal{S})$ is to force the learned state codes of $\mathcal{S}$ and $\mathcal{T}$ to express the data correlations subject to local flow conservation. Inspired by Contrastive Predictive Coding (CPC) (Oord et al., 2019), we learn the mappings from $\mathcal{S}$ to $\mathcal{T}$ in a way that maximally preserves the mutual information of the source and target sequences defined as follows:

$$I(\mathcal{T}; c) = \sum_{\mathcal{T}, c} p(\mathcal{T}, c) \log \frac{p(\mathcal{T}|c)}{p(\mathcal{T})} \tag{1}$$

where $p(\cdot)$ is the probability function.

In analogy to CPC, we rely on a contrastive loss function to train the model. As the practice of contrastive learning (Chen et al., 2020; Oord et al., 2019), we first construct a training batch of $N_b$ sequence samples, $\Phi = \{\mathcal{T}_1^\star, \mathcal{T}_2, \cdots, \mathcal{T}_{N_b}\}$. $\mathcal{T}_1^\star$ is the positive sample from the conditional distribution $p(\mathcal{T}|c)$ for the target sequence $\mathcal{T}$ and other $N_b - 1$ sequences are the negative samples from the original distribution $p(\mathcal{T})$. Accordingly, we define the following contrastive induction loss $\mathcal{L}_I$ to optimize the mutual information in Equation 1 as follows:

$$\mathcal{L}_I = -\mathbb{E}_\Phi \left[ \log \frac{f(\mathcal{T}_1^\star, c)}{\sum_{\mathcal{T}_j \in \Phi} f(\mathcal{T}_j, c)} \right] \tag{2}$$

where $f(\cdot)$ is a score function proportional to the ratio $p(\mathcal{T}|c)/p(\mathcal{T})$. As the way done in CPC, this can be technically implemented by using a simple log-bilinear model:

$$f(\mathcal{T}_j, c) = \sum_{t \in T_2} \exp(trans[h_{\mathcal{T}_j}^t] \cdot \hat{h}_{\mathcal{T}_1^\star}^t) \tag{3}$$

where $trans[h_{\mathcal{T}}^t]$ is the transpose of the state coding $h_{\mathcal{T}}^t$ from the initial embedding layer in Fig. 4.

The induction loss $\mathcal{L}_I$ defines a categorical cross-entropy of classifying the positive sample correctly. Optimizing $\mathcal{L}_I$ will result in high similarity between the true embedding codes $h_{\mathcal{T}_1^\star}^t$ and the induced state codes $\hat{h}_{\mathcal{T}_1^\star}^t$. The above induction pattern is learned by directly comparing the target coding with the codings of randomly sampled negative samples in latent space in a self-supervised manner. Such process requires no direct labels in the output space, which makes it easier to generalize the induced knowledge to diverse applications in IP networks.

## 4 EXPERIMENTS

We validate the efficiencies of *FlowNN* on two crucial networking application tasks in IP networks–flow rate prediction and end-to-end delay inference.

**Datasets.** We conduct experiments on two publicly available flow datasets–WIDE and *NumFabric*[7]. WIDE are the realistic traffic traces collected from real-world network environment, and *NumFabric* are the synthetic traffic widely applied in recent networking researches (Nagaraj et al., 2016; Lei et al., 2019). The network topology includes 4 core nodes, 8 leaf nodes and $8 \times 16$ hosts organized in accordance with Fat-Tree network architecture (Al-Fares et al., 2008). We sampled the flow rates of 50 flows as well as the associated aggregated flow rates at each nodes at the time scale of $1ms$ for total length $18432ms$. Each flow traverses a routing path with 5 nodes. We found that the flow tensor time series of such length are enough to learn a stable flow pattern at a sub-second granularity.

**Baselines.** We compare the proposed *FlowNN* with the following baselines: 1) Seasonal Auto-Regression Integrated Moving Average (SARIMA) (Hyndman & Khandakar, 2008; Wang et al., 2006); 2) GRU (Chung et al., 2014); 3) multiple GRUs (multiGRU); 4) STHGCN (Kalander et al., 2020), and 5) STGCN (Yu et al., 2018). Particularly, GRU encodes and predicts each time-series sequence stand-alone without any reference to the information from other spatial nodes. In contrast, multiGRU uses separate GRUs to encode each time-series sequence but predicts with Fully-connected Neural Network (FNN) by jointly concatenating the codings of sequences from all spatial nodes. STHGCN is the recent work for networking traffic analysis, which uses graph convolutional

---

[7]The data can be generated by running the simulation code released in `https://knagaraj@bitbucket.org/knagaraj/numfabric.git`

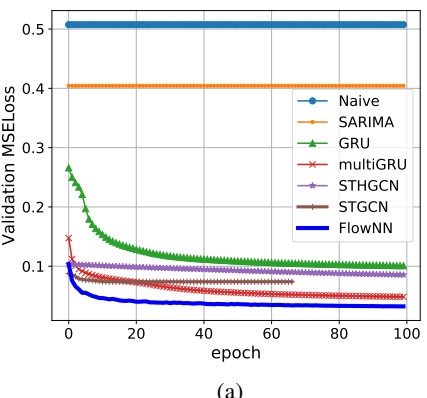 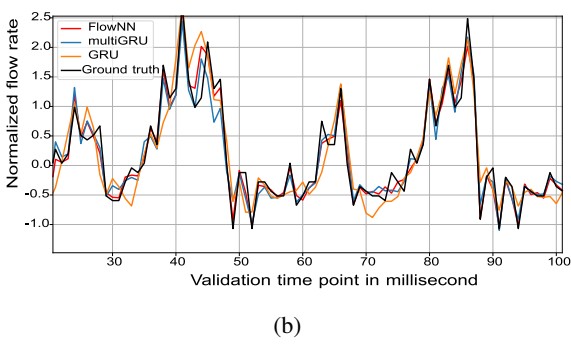

(a)

(b)

Figure 5: (a) Validation MSELoss; (b) Normalized flow rate predicted by different models.

network (Kipf & Welling, 2016) and GRU to predict the spatial states of all nodes at each time. Finally, STGCN is a widely applied model for transportation traffic analysis, which is build upon the spatio-temporal graph convolution.

**Implementation Details.** We use Adam (Kingma & Ba, 2014) with a learning rate of $1e^{-4}$. All hidden dimensions are set to 128 and the layer size for GRUs is 3. We train on sampled time series window of length 256. A batch of $N_b = 10$ is used to draw the samples for the contrastive induction loss in Equation 2. The initial embedding layer in *FlowNN* is performed for each node data by separated FNNs. We first pre-train the *FlowNN* with the contrastive induction loss until convergence and then fine-tune the model with MSE loss (Equation 4) for different application tasks.

**Evaluation metrics.** The following metrics are adopted for evaluating quality of prediction $\hat{y}$ with the ground truth $y$.

$$\text{Mean squared error:} \qquad MSE = \frac{1}{n} \sum_{i=1}^{n} (y_i - \hat{y}_i)^2 \qquad (4)$$

$$\text{Relative absolute error:} \qquad RAE = \frac{1}{n} \sum_{i=1}^{n} \mid \frac{y_i - \hat{y}_i}{y_i} \mid \qquad (5)$$

$$\text{Correlation coefficient:} \qquad Corr = \frac{\sum_{i=1}^{n} (y_i - \bar{y})(\hat{y}_i - \bar{\hat{y}})}{\sqrt{\sum_{i=1}^{n} (y_i - \bar{y}_i)^2 \sum_{i=1}^{n} (\hat{y}_i - \bar{\hat{y}})^2}} \qquad (6)$$

where $\bar{y}$ and $\bar{\hat{y}}$ are the mean values of $y$ and $\hat{y}$.

### 4.1 PATH-WIDE VIEWS MATTER FOR IP TRAFFIC FLOW MODELLING!

We first use the one-step-ahead prediction task to demonstrate that path-wide views of *FlowNN* are more informative than the widely used single-node view or graph-level views in the literature.

Fig. 5a shows the validation MSE loss across training epochs for the one-step-ahead prediction task. *Naive* solution directly uses the previous observation as the next-step prediction. The results of *Naive* and SARIMA provide a good testament to the prediction difficulty for traditional non-DL solutions in a millisecond-granularity networking prediction task. We can see that the loss of SARIMA is still as high as 80% of the *Naive* solution. However, the losses of all DL based solutions are less than 20% of *Naive*. This shows the powerful advantages of the deep learning principle in difficult tasks. Table 1 shows the test performances of different methods. Specifically, GRU holds a single node view and learns the time series of each node separately and the achieved loss is only 22.5% of SARIMA. By contrast, the other DL solutions harvest the information from all path nodes to predict (i.e., the path-wide views) and the loss are remarkably reduced further against GRU. As discussed in **Path-level Coding**, STGCN and STHGCN inherit the graph aggregation operation,

Table 1: Test performance comparison of different models on one-step-ahead prediction task

| Model | NumFabric | | | WIDE | | |
|---|---|---|---|---|---|---|
| | MSE | RAE | Corr | MSE | RAE | Corr |
| Naive | 0.5074 | 1.6818 | 0.7462 | 1.0017 | 5.7229 | 0.2619 |
| SARIMA | 0.4043 | 1.5122 | 0.7924 | 0.7164 | 4.4304 | 0.2979 |
| GRU | 0.0911 | 1.3441 | 0.9514 | 0.4634 | 3.8070 | 0.7038 |
| multiGRU | 0.0409 | 0.9406 | 0.9784 | 0.3972 | 3.7637 | 0.7532 |
| STGCN | 0.0738 | 0.8510 | 0.9608 | 0.4351 | 2.9718 | 0.6791 |
| STHGCN | 0.0716 | 0.8930 | 0.9587 | 0.4341 | 3.2210 | 0.5620 |
| *FlowNN* | **0.0297** | **0.7960** | **0.9843** | **0.3523** | **3.6223** | **0.7703** |

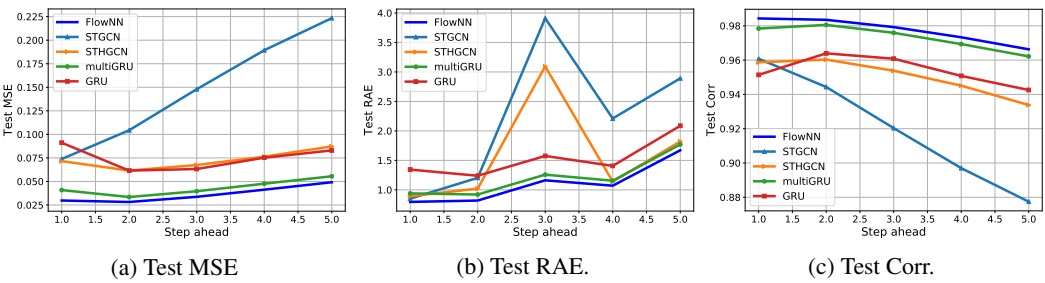

| (a) Test MSE | (b) Test RAE. | (c) Test Corr. |
|---|---|---|

Figure 6: Test performances of different DL models on multi-step-ahead predictions task.

their performances are better than GRU, but are inferior to multiGRU. Such findings discourage the spatial aggregation operation in GCN as the data correlation exploited by GCN deviates a lot from the truly S-shaped correlation in IP flow data. Finally, the proposed *FlowNN* outperforms all the baselines.

Fig. 5b compares the predictions by different models against the ground truth. We can observe that *FlowNN* and multiGRU capture the trend of ground truth accurately. Benefiting from the *Induction Operation*, *FlowNN* avoids over-reliance on the far distant history as the vanilla GRU and its variants function and shows the fastest response to the dynamic changes among the traffic flows.

## 4.2 WIDE APPLICABILITY ON DIFFERENT APPLICATION TASKS

**Multi-step-ahead predictions**: We first exam the effectiveness of *FlowNN* on the multi-step-ahead prediction task. In this task, we first pretrain *FlowNN* based on the contrastive induction loss in Equation 2 until convergence. Following the practice of the recent work for the short-term (second-level granularity) traffic forecasting in (Mozo et al., 2018), we then take the output of pre-trained *FlowNN* and finetune a FNN based Readout layer in Fig. 4 to infer the mean of multi-step-ahead predictions. The mean realizations of the multi-step ahead are crucial to many short-term networking planning and control tasks in IP networks. Fig. 6 compares the test performances of different DL models on the *NumFabric* dataset. The test results shows that *FlowNN* outperforms all these recent baselines on all metrics.

**End-to-end delay inference**: In today's IP network, it is important to guarantee that the provided QoS meets the subscribed service agreement, such as end-to-end transmission delay less than 50ms for online gaming. However, it is difficult to build an accurate delay model for human experts (Xiao et al., 2018; Rusek et al., 2019) in practical networks since many factors may influence the delay, including the dynamic traffic demands, packet processing and queueing etc. In this task, we apply the learned traffic flow model from *FlowNN* to perform the traffic-driven delay inference. Specifically, we apply the same *FlowNN* model pretrained as in multi-step-ahead prediction task, and finetune a new FNN based Readout layer to infer the next-step delay. Table 2 shows the test performances of different models. Although pretrained on the dataset of traffic flow rates, *FlowNN* still achieves the best results on the task of inferring the data with different physical meaning. This shows the robustness of *FlowNN* across different tasks.

Table 2: Test performance comparison of different models on end-to-end delay inference

| Model | MSE | RAE | Corr |
|--------|--------|--------|--------|
| GRU | 0.1364 | 1.8333 | 0.9300 |
| multiGRU | 0.1011 | 1.4008 | 0.9466 |
| STGCN | 0.1263 | 2.0247 | 0.9412 |
| STHGCN | 0.0922 | 1.4683 | 0.9533 |
| *FlowNN* | **0.0724** | **1.3570** | **0.9617** |

Table 3: One-step-ahead prediction performances of different models in Out-Of-Distribution test. For *Same dataset test*, both the model training and test are conducted on WIDE. By contrast, in *Cross dataset test*, models are first trained on *NumFabric*, but then test on WIDE by finetuning their Readout layers.

| Model | Same dataset test | | | Cross dataset test | | |
|--------|--------|--------|--------|--------|--------|--------|
| | MSE | RAE | Corr | MSE | RAE | Corr |
| multiGRU | 0.3972 | 3.7637 | 0.7532 | 0.4272 | 3.7717 | 0.7126 |
| STHGCN | 0.4341 | 3.2210 | 0.5620 | 0.5236 | 3.3330 | 0.4715 |
| *FlowNN* | **0.3523** | **3.6223** | **0.7703** | **0.4173** | **3.9001** | **0.7213** |

### 4.3 GENERALITY TEST ON OUT-OF-DISTRIBUTION DATASET

In this subsection, we test the model generality when a *FlowNN* model, pretrained with the contrastive induction loss on one dataset, say *NumFabric*, is used to work in an environment that is different from the trained one. This is performed by testing with the one-step-ahead prediction task on *Out-Of-Distribution* dataset (i.e., cross dataset test). Specifically, we take the *FlowNN* model pretrained on the *NumFabric* dataset and finetune the FNN based Readout layer to test its prediction performances on the WIDE dataset. As comparison, we also finetune the Readout layers of multi-GRU and STHGCN already trained on *NumFabric* to test their performances on WIDE. From the test results shown in Table 3, we can observe that *FlowNN* achieves the best performances except that RAE is slightly higher. Moreover, the performances of *FlowNN* achieved in cross dataset test is even better than the results of other baselines achieved in same dataset test (except multiGRU). This shows the good generality and robustness of *FlowNN*.

**Discussions**: We observe that multiGRU, to certain extent, also works well in above experiments, although is inferior to *FlowNN*. This can be interpreted by the shortcut learning, as disclosed in (Geirhos et al., 2020). As illustrated by the data correlation in Section 2, the near future realization of a node is highly correlated to the path-wide values at last step. This makes a shortcut for multiGRU that a prediction by directly weighting the last-step path-wide value will capture such kind of data correlation, although this can not further extract the effect of local flow conservation along the routing path as *FlowNN* functions.

## 5 CONCLUSION

In this paper, we formulated the crucial traffic flow modelling problem in IP networks, and develop a customized neural network–*FlowNN* and the associated *Induction Operation* to extract the domain-specific spatio-temporal data correlations in IP traffic flows. This study makes the pioneering work to design customized neural network and learning strategy by deeply combining the IP network structure and working mechanism. We reported the state-of-the-art performances for multiple practical networking application tasks, which demonstrates the strength of our approach. As a new 'playground' for both networking and deep learning communities, the research of network intelligence is still in its infancy. We hope this work will inspire more innovations in this field in future.

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
