# OpenReview forum: "Flow Neural Network for Traffic Flow Modelling in IP Networks"
_ICLR.cc/2021/Conference — Reject_

### Official Review · AnonReviewer3 · 2020-10-28
**Possibly an interesting approach but difficult to follow due to lack of clarity.**

**Rating:** 3
**Confidence:** 2

**Review:**

The work proposes a new way to analyse flow structured data using what they call flow neural networks, which supposedly better exploits correlations between different connected nodes at different time-points. The approach is tested on the public dataset NumFabric with better results than the compared to approaches.

Although I believe I get the rough outline of what the authors are proposing, there is simply too many unclear aspects of the work for it to be published at ICLR. Prior work, especially the benchmarked methods, also seems to be mostly tangentially related, e.g. covering general methods for time-series or graph analysis and not specific methods for traffic analysis.

Quality
A single dataset is used for comparison and it is unclear how relevant the baselines are, as there appears to be a lot of prior work on deep learning for traffic analysis, some of this is cited, but not benchmarked against. See also the additional references [1, 2], and the large number of works citing them. It would be good if the authors could explain why their chosen set of baselines are relevant.

Clarity
The work is hard to understand. An incomplete list of things follows below.

- Crucially I don't clearly understand the STI effect as explained in definition 1.
- Why do you specifically mention self-driving Tesla cars, wouldn't this be relevant for other self-driving cars as well?
- What do you mean by more than 1 million flows, that is, how do you count flows?
- "As the advent of many advanced machine learning (particular deep learning), intelligent flow analysis tools are gaining more momentum to proceed", sentence does not make sense.
- I do not understand the paragraph starting with "Additionally, in contrast to the learning analysis of formation-agnostic natural objects, ..." What is a formation-agnostic natural object? How can a network system enjoy rich domain expertise? How can network traffic experience something? Etc.
- temproal -> temporal
- was originated -> originated
- "This essentially discriminates", does not really make sense to me in this context.
- Sentence starting with "In the path-constrained propagation process as doing in above IP traffic flows,..", does not make sense to me.
- What is APP?
- It would be good to use the same number of decimal points for the result of each method in Table 1.
- "However, all deep learning based solutions achieve several times to tens times better accuracy.", I do not see this in the figure.
- Please use consistent naming schemes, see "uniGRU" and "univarGRU".
- In summarizing the results (4.1), I think the word improves is a bit unclear when talking about reduction in loss. E.g. the loss is only xx percent of yy would be more clear.
- "This is benefited from the STI effect" -> "This is benefiting from the STI effect"

Originality
The proposed work may very well be original and novel. I am just not convinced.

Significance
In its present form, I doubt it will have much influence.

[1] Polson, Nicholas G., and Vadim O. Sokolov. "Deep learning for short-term traffic flow prediction." Transportation Research Part C: Emerging Technologies 79 (2017): 1-17.
[2] Cui, Zhiyong, et al. "Traffic graph convolutional recurrent neural network: A deep learning framework for network-scale traffic learning and forecasting." IEEE Transactions on Intelligent Transportation Systems (2019).

---

### Official Review · AnonReviewer1 · 2020-10-28
**This paper proposes a Flow Neural Network (FlowNN) to model the flow-structured data. However, the writing and experiments should be improved.**

**Rating:** 4
**Confidence:** 3

**Review:**

In this paper, the authors present a new Flow Neural Network (FlowNN) to model the flow-structured data. The main idea is to employ the unique properties of the network traffic flows, i.e., flowing invariance and variance.

Overall this seems like a nice attempt to model the flow-structured data. However, the paper should be improved in the following aspects.

- Presentation: My first concern is the writing of this paper.
1) After reading the first two sections, I was really confused about what the problem is. According to the experiment part, this paper seems to study the flow prediction task. What is the difference between this problem and spatio-temporal prediction? I strongly suggest that the authors write a section for introducing the problem statement.
2) What is flow data? There seems no mathematical formulation of the network flow data (seemingly a tensor).
3) More related works (e.g., flow-structure data mining, spatio-temporal GCNs, spatio-temporal forecasting) should be included to help the audiences better understand the context.

- Experiments:
1) The baselines are weak. To the best of my knowledge, in the recent 2 years, there are many papers [1, 2, 3] published at top-tier conferences (e.g., KDD, IJCAI, AAAI) that achieve much better results than the baselines included in this study. Why not comparing FlowNN with them?
2) The proposed method is only evaluated on a single dataset, which limits its universality. If possible, more datasets should be considered.
3) How about the robustness and stability of the proposed method? The variance of the performance should be present.
4) In real-world applications such as traffic forecasting, multi-step ahead forecasting is more practical than 1-step prediction. It would be good to evaluate the effectiveness when predicting more future horizons.

- Minor issues:
1) The reference of GCN-GRU is wrong. As I know, Yu et al. 2018 combined temporal convolutional networks with GCN, instead of using RNN for capturing temporal dependencies.
2) what is the meaning of the S-shaped road in Fig. 1?

Reference:
[1] Wu et al., Graph WaveNet for Deep Spatial-Temporal Graph Modeling, IJCAI 2019
[2] Guo et al., Attention Based Spatial-Temporal Graph Convolutional Networks for Traffic Flow Forecasting, AAAI 2019
[3] Pan et al., Urban Traffic Prediction from Spatio-Temporal Data Using Deep Meta Learning, KDD 2019

---

### Official Review · AnonReviewer4 · 2020-10-28
**The paper raises some valid points but falls short in providing consistent analysis and proof for their claims. The experimental setup needs to be improved and include general flow network types and stronger benchmarking.**

**Rating:** 4
**Confidence:** 4

**Review:**

Summary:

The goal of this study is the 1-step prediction of flow rate in flow networks. They first define a “spatial-temporal induction effect (STI)” and claim it to be the universal property of flow networks. Their main contribution is their proposed “flow neural network” which is based on the STI effect and a combination of GCN and GRU architectures.  According to authors, the novelty of their work lies in the fact that they consider the spatiotemporal features of the flow network simultaneously, whereas the previous works only consider them separately.

%%%%%%%%%%%%%%%%%%%%%%%%%%%%%%%%%%%%%%%%%%%%%%%%%%

Strengths of the paper:

The raised concern over the fact that considering the time series of the node alone does not capture the full complexity of the flow dynamics is valid and an interesting direction to pursue.

Their proposed model achieves a better validation loss than the benchmarks

The paper does a decent job in pointing out the flaws of the previous models.

%%%%%%%%%%%%%%%%%%%%%%%%%%%%%%%%%%%%%%%%%%%%%%%%%%

Detailed Review:

Major concerns:

For a paper that heavily relies on empirical results (as mentioned throughout the paper), there is a serious lack of various experimentations to prove the followings:

- The claimed universality of spatiotemporal induction (STI) effect,

- Considering the spatial-temporal features separately (i.e., spatial-spatial and temporal-temporal) indeed results in considerably poorer performance in different settings,

- The practical benefit of the proposed architecture tested on different real-world networks from different domains (to comply with the title and the claim in the paper that the proposed method is extensible to other flow networks besides IP networks, as well).

The authors claim the inefficiency of previous studies (some of which are based on strong mathematical theories) on the same domain (e.g., Harchol-Balter, 2013; Ciucu & Schmitt, 2012; Ciucu et al., 2019) without proof and do not consider these works in their baselines. In fact, many of the selected baseline belong to relatively old studies that are not necessarily SOTA for the flow prediction task right now. There are numerous recent studies on network flow prediction (e.g., traffic flow, crowd flow, etc.) combing the spatial-temporal features of the system through (e.g., these two papers to name a few:

https://www.sciencedirect.com/science/article/pii/S0968090X19301330?casa_token=_HkqrohsGwEAAAAA:_qI_xcFnpiFkXGzqxhXNTfEx4jJA_EoAN673pBimQezdlIFCw_79EWTcB9XIhpSiCGwhJ9W_Pyx9

https://www.aaai.org/ojs/index.php/AAAI/article/view/3892)

I suggest authors develop their related work section further and use more SOTA baselines in their experiments. Also, choose more than one dataset from different domains for validating the proposed model.

%%%%%%%%%%%%%

The definition of “spatial temporal induction” is rather vague and does not clarify what STI effect means, both theoretically and intuitively. There are no proof on the “universality” of this effect for all flow networks and the two properties listed in page 4 are neither proven nor connected to the rest of the paper. What is the significance of STI and why do we care about its two properties (if they indeed hold).

%%%%%%%%%%%%%

The paper is rather tough to read. There are instances of using abbreviations before they are defined and mathematical notations with variables that are not properly defined (ref. Page 5).

%%%%%%%%%%%%%

How are the flow timeseries defined as (in-flow? Out-flow?). On the same note, how does the ordering of the subtraction affect figure 4.b (e.g. changing 130-136 to 136-130 while keeping the rest of subtractions the same).

%%%%%%%%%%%%%

The figures in general are not well connected to the text. Majority of them include details that are not explained in the text (or the caption) and a number of them are hard to interpret. For example,

Figure 1.b: what is the red box for?

Figure 3.a: what is x_ijt in terms of flow rate (I can only attribute it to in-flow and out-flow which does not seem to be used in this study)? And what does X_iit imply? How is the blue curve (called S curve) obtained in the figure?

Figure 3.b: what is to be inferred from this figure?

Figure 4.a and 4.b: Are these based on real data or they are only toy examples? If it is the latter, then it cannot be used to prove the existence of STI effect. If it is the former, what is the data? It is hard to attribute meaning to the graphs without knowing the context of the data.

 Moreover, in figure 4.a, shouldn’t the spike in the source node 41 reach to neighboring node 130 with some time delay (the same for other nodes as well)?

Figure 5: Again, what is X_ij?  What is the direct output from “flowing invariant coding” used for?

Figure 6: I find it hard to interpret the caption from the figure.

Figure 8: what is to be inferred from the red boxes?

I suggest authors increase the font and quality of their figures and enrich their captions to convey the intended message with more clarity.

%%%%%%%%%%%%%

The problem statement and novelty of the study is not clear from the introduction section.  I suggest authors clearly define the problem and point out the novelty of their work in comparison with current studies from the beginning.

%%%%%%%%%%%%%

Please elaborate on how the flow structured data is a “radically new data type” (ref. Section 1)

%%%%%%%%%%%%%

The results in figure 7 are called “validation” loss. Does that imply these nodes have been included in the training or hyperparameter tuning of the model? If yes, it is more interesting to know how model performs for flow prediction of unseen nodes.

%%%%%%%%%%%%%

It is interesting to know how the flow prediction task for K-step ahead will be.

%%%%%%%%%%%%% %%%%%%%%%%%%%

Minor Concerns:

The paper has to be self-sufficient, so for equation 1 and 2, more explanation is needed instead of referring readers to the previous work.

%%%%%%%%%%%%%

Some typos and grammatical errors that need to be fixed. Some examples:

Section 1: “tens to tens thousand of nodes”, “particular deep learning”

Figure 3 caption: repeated “the”

Section 2: “any pair of nodes in the routing path will become larger interchangeably than the other so that the flux is conserved among nodes”

Section 3: “a encoding”, “a RNN”, “While any encoder and decoder can be used so long as we can backpropagate through it.”, “Analogy to CPC, we define the following induction loss...”

Page 6: the two lines before section 4 do not seem to connect correctly with the previous page.

 %%%%%%%%%%%%%%%%%%%%%%%%%%%%%%%%%%%%%%%%%%%%%%%%%%

Questions:

Please answer the questions raised in “detailed review” above.

---

### Official Review · AnonReviewer2 · 2020-10-29
**The paper cannot be well understood and evaluated**

**Rating:** 2
**Confidence:** 4

**Review:**

Summary:
- The paper claims to discover a universal “spatio-temporal induction (STI) effect” in network traffic flows, and developed a model FlowNN to learn representations of flow-structure data. However, the STI effect was not clearly explained and the problem is not well formulated, making it hard for readers to understand the value of this work.

Strong points:
- Modeling flow-structure data and internet traffic flows is an interesting problem.
- The paper motivates the problem well.

Weak points:
- The writing of this paper is very confusing and unclear.
- The problem was not clearly formulated or articulated.
- The descriptions of the model and STI effect are confusing.

Recommendation:
- I strongly recommend a reject. While the topic is interesting, the paper is hardly understandable in its current form. The contributions cannot be well understood and evaluated. Improving clarity will make it a much stronger submission in the future.

Comments & questions:
- What exactly is the task? It was never clearly stated in the paper. Only mentioning “Learning the representations” or “regression tasks” is unclear and confusing.
- What is the training and test data split? How does the approach generalize to unseen networks or flows?
- The definition of STI is unclear. In Definition 1, the paper writes “Spatio-Temporal Induction defines the synchronized map process to produce the future/spatial evolution between neighboring nodes.” This does not define the STI effect. What exactly is the “synchronized map process”? Different network flows process packets at different paces, so how are they synchronized?
- How can the representation (Fig 3a) scale with the number of nodes in the network? You would need one such 3D matrix for a flow. The paper evaluates with 28 nodes in total (Sec. 4), but the real internet is orders of magnitude larger (billions and millions of flows).
- The meaning of Fig 4. is unclear. “a spike at a upstream node in Fig. 4a spurts the flow at downstream nodes in certain subsequent time windows.” However, it’s very hard to see that from the figure. It seems like all flows are fluctuating at different time steps.
- Many claims seem to be wrong or not supported. A few examples:
  - “a encoding net with shared weights is learnt to extract the common patterns shared in all received inputs.” Share weights imply extracting “similar” patterns across flows, how can the design extract “common” patterns?
  - “This reconstructs the evolution patterns of the APP source flows that drive the evolution processes at all nodes along the path.” -> How exactly?
- The meaning of many sentences are unclear. A few examples:
  - “the flow rates be- tween two neighbouring or any pair of nodes in the routing path will become larger interchangeably than the other so that the flux is conserved among nodes.”
  - “This not only experimentally confirms the S-shaped spatio-temporal correlation present in the flow-structured data, but also further confines the correlation with the explicit (partial) flow conservation.”
  - “Induction is operated on … by the paired induction operator.”
- Many grammar issues. A few examples:
  - “Data can be organized compact and complete”
  - “propagation process as doing in above IP traffic flows”
  - “While any encoder and decoder can be used so long as we can backpropagate through it.”

Suggestions
- Writing-wise, it will be helpful to simplify the wording and be direct and specific.
- It’ll be helpful to ask people not familiar with the project to read the paper and get feedback.

==== Updates after the response ====

I appreciate the authors’ effort in updating the manuscript. The new manuscript has significant changes, but still many questions are left unanswered. Thus, I’m keeping my rating and recommendation.

---

### Author Response · Authors · 2020-11-25
**revision summary for the provided response letter in supplementary material**

We appreciate the time and efforts all reviewers and ACs have devoted to the review of this paper. Your comments and suggestions help us a lot to improve the quality of this paper.  Basically, all reviewers posted many shared concerns about the writing and vague problem formulation in our previous submission that make all reviewers hard to understand and evaluate. With the help of the suggestions and questions all reviewers provided, we have reorganized the writing of this paper and added more details to illustrate the IP networking backgrounds, working mechanism, the motivation of our problem and methods, as well as more experiments to justify the novelty, superiority and robustness of our proposal.  Please see the response letter attached in Supplementary Material for more details.

---

### Decision · Program_Chairs · 2021-01-07
**Final Decision**

**Decision:**

Reject

**Comment:**

The revised paper is a solid improvement.  However, all reviewers and I find that there are still a number of issues that prevent the paper from being acceptable at the current stage.  For example, some important parts are still unclear, especially the definition of STI effect.  The observation of STI effect requires more theoretical or empirical investigation, in addition to a toy example.